# Effect of the Ankle–Foot Orthosis Dorsiflexion Angle on Gait Kinematics in Individuals with Hemiparetic Stroke

**DOI:** 10.3390/bioengineering12101091

**Published:** 2025-10-10

**Authors:** Hiroshi Hosokawa, Fumiaki Tamiya, Ren Fujii, Ryu Ishimoto, Masahiko Mukaino, Yohei Otaka

**Affiliations:** 1Department of Rehabilitation Medicine, Yamaga Hot Spring Rehabilitation Hospital (Mokuseikai Medical Corp.), Kumamoto 861-0514, Japan; hiroshi.hosokawa@fujita-hu.ac.jp; 2Musashigaoka Clinical Research Center, Musashigaoka Hospital (Tanakakai Medical Corp.), Kumamoto 861-8003, Japanr-fujii@tanakakai.com (R.F.); 3Department of Rehabilitation Medicine, School of Medicine, Fujita Health University, Toyoake 470-1192, Japan; 4Department of Rehabilitation, Musashigaoka Hospital (Tanakakai Medical Corp.), Kumamoto 861-8003, Japan; 5Department of Rehabilitation Medicine, Hokkaido University Hospital, Sapporo 060-8648, Japan

**Keywords:** stroke, orthotic devices, ankle-foot orthoses, gait analysis

## Abstract

Ankle-foot orthoses (AFOs) are widely used to improve gait; nonetheless, it remains unclear how specific settings, particularly the dorsiflexion angle, affect gait kinematics in individuals with stroke. This study investigated the effect of different AFO dorsiflexion angles on gait kinematics in ambulatory adults with hemiparesis. Twenty-six individuals with post-stroke hemiparesis walked on a treadmill while wearing the same type of AFO at four ankle dorsiflexion angles: 0°, 5°, 10°, and 15°. Temporal-spatial variables, joint angles, and toe clearance and its components were quantified using three-dimensional analysis. The double-stance time before the paretic swing shortened significantly with increasing dorsiflexion angle, whereas the mean stride time and length did not significantly change. During the swing phase, increased AFO dorsiflexion was associated with reduced maximal knee flexion, in addition to its direct effect on ankle angles. The absolute toe clearance height was unaffected by the AFO settings; however, the contribution of ankle dorsiflexion to limb shortening increased stepwise from 0° to 15°, and the hip elevation and compensatory movement ratio declined. In conclusion, increasing the AFO dorsiflexion angle significantly altered gait kinematics, with distal ankle mechanics replacing inefficient hip compensation and reducing double-stance time.

## 1. Introduction

Recovery of walking function is one of the primary goals in individuals with stroke, and the identification of effective and efficient approaches to restoring post-stroke gait is crucial for stroke rehabilitation [1]. Individuals with hemiparetic stroke have been previously reported to often exhibit specific gait abnormalities resulting from a combination of functional impairments and compensatory movements. For example, during the swing phase, functional impairments reduce ankle dorsiflexion, hip flexion, and knee flexion, which negatively affect toe clearance [2,3,4,5,6,7]. Induced reduction in knee flexion may necessitate compensatory movements, such as hip elevation and circumduction, to achieve toe clearance [4,8,9]. Although these compensatory movements are often indispensable for achieving gait in individuals with motor impairment, they are associated with increased energy expenditure and could be related to fatigue or long-term musculoskeletal strain [4,10]. Thus, caution should be exercised to avoid an over-reliance on these compensatory movements, considering their potential drawbacks.

In clinical practice, ankle-foot orthoses (AFOs) are commonly prescribed to address gait abnormalities, including reduced toe clearance and insufficient stance stability. Although prior studies have shown that AFOs can improve toe clearance and reduce compensatory movements such as hip elevation in stroke survivors [11,12,13,14], most of this research has compared walking with and without an AFO rather than systematically adjusting device settings. For clinical application, however, a clearer understanding of the dose–response relationship is necessary to guide these adjustments, as toe clearance is highly sensitive to ankle angle [15,16], and current practice remains largely empirical. Although graded angle adjustments have been shown to systematically alter toe clearance control in healthy adults [17], it remains unclear whether these principles apply to individuals with stroke.

The present study was designed to address this knowledge gap. To our knowledge, it is the first to systematically grade the AFO dorsiflexion angle across multiple settings in a stroke cohort to quantify the resulting dose–response effects. Our primary objective was to evaluate the effects of varying AFO dorsiflexion angles on gait patterns in individuals with mild hemiparesis, with a particular focus on swing-phase kinematics and toe clearance control, which are often the most critical challenges in this population. We hypothesized that increasing the AFO dorsiflexion angle would enhance ankle-driven limb shortening and reduce reliance on proximal compensations. To test this hypothesis, three-dimensional motion analysis was conducted to quantify the kinematics, and vertical components contributing to toe clearance and compensatory movements were calculated. Understanding the impact of AFO angle adjustments on gait abnormalities may provide valuable insights into optimal rehabilitation strategies and individualized AFO settings for individuals with hemiparetic stroke.

## 2. Materials and Methods

### 2.1. Study Design and Setting

This experimental study was conducted at Musashigaoka Hospital (Kumamoto, Japan). The study protocol was approved by the Medical Research Ethics Review Committee of Fujita Health University (approval no.: HM22-309, approval date: 14 February 2023). The study was conducted in accordance with the principles of the Declaration of Helsinki (1964), as revised in 2013. All participants provided written informed consent prior to study enrollment.

### 2.2. Participants

Individuals with hemiparetic stroke who were undergoing rehabilitation at Musashigaoka Hospital from March 2023 to August 2024 were recruited via convenience sampling. The inclusion criteria were as follows: participants who (i) were diagnosed with hemiparetic stroke, (ii) were undergoing rehabilitation at Musashigaoka Hospital (either as inpatients or outpatients), (iii) could walk independently, (iv) used a prescribed AFO, (v) could walk steadily on a treadmill without handrail support, and (vi) provided written informed consent. The exclusion criteria were as follows: participants who (i) were unable to walk on a treadmill for 20 s and (ii) had consciousness disorders or cognitive impairments that rendered following test instructions difficult. A priori power analysis was conducted using G*Power software version 3.1.9.3 [18] to determine the sample size. With a presumed medium effect size (*f* = 0.25), significance level (α) of 0.05, and statistical power (1 − β) of 0.80, the analysis indicated a required sample size of 24 participants. To account for potential attrition, the final sample size was set at 26.

### 2.3. Experimental Setup and Conditions

Gait analysis was performed using a three-dimensional treadmill gait analysis system (KinemaTracer; Kissei Comtec Co., Ltd., Matsumoto, Japan). This system consisted of a data-recording PC and four charge-coupled device cameras operating at 60 Hz, which were positioned around the treadmill (Ohtake-Root Kogyo Co., Ltd., Ichinoseki, Japan) to capture three-dimensional marker coordinates. Markers were affixed to anatomical landmarks, including the bilateral acromions, hip joints (one-third from the greater trochanter along a line connecting the anterior superior iliac spine and the greater trochanter), knees (at the midline of the anteroposterior diameter of each lateral epicondyle of the femur), ankles (at the lateral malleoli), toes (at the fifth metatarsal head), and iliac crests (positioned vertically in line with the hip joints), totaling 12 markers (Figure 1). The reliability and validity of measurements using KinemaTracer have been reported previously [14,19].

The AFOs used in this study were adjustable posterior strut AFOs (APS-AFOs; Tomei Brace Co., Ltd., Seto, Japan), which are articulated devices commonly utilized for gait rehabilitation in Japan [20,21,22]. These AFOs allow for customization of the ankle hinge joint and strut properties. In this study, a hard aluminum strut was used, and the APS-AFO was applied to the hemiparetic leg. Leg length discrepancies were measured and corrected using insoles. Measurements were performed at four fixed ankle dorsiflexion angles with the AFO: 0°, 5°, 10°, and 15° (Figure 2). To minimize bias, the order of measurements was randomized, and participants rested for 5–10 min between sessions. The treadmill walking speed was set to 70% of each participant’s self-selected comfortable speed, which was determined through a 10-m walk test. Data were collected over a 20-s duration for each condition.

### 2.4. Kinematic Analysis

Gait cycle events, specifically heel strike and toe-off, were determined algorithmically using the trajectories of the ankle and toe markers within KinemaTracer (Kissei Comtec Co., Ltd.). The spatiotemporal parameters were computed for both limbs. Temporal parameters included the stride time, paretic swing time, paretic single-stance time, and double-stance time before and after the paretic swing. Spatial metrics included the stride length and paretic/non-paretic step length.

The hip, knee, and ankle joint range of motion was quantified during the stance and swing phases of the gait cycle. This involved calculating the peak joint angles for hip flexion/extension, knee flexion/extension, and ankle dorsiflexion/plantar flexion during each phase. The hip, knee, and ankle joint angles were calculated in the sagittal plane using the marker position data. The hip angle, represented by the trunk-thigh angle, was defined by the positions of the iliac crest, hip, and knee markers. The knee angle (i.e., the thigh-crus angle) was determined using hip, knee, and ankle markers. Finally, the ankle angle (i.e., the crus-foot angle) was computed using knee, ankle, and toe markers.

Toe clearance of the paretic limb during the swing phase and its contributing components were quantified by adhering to procedures established in prior research [15,23,24,25]. Toe clearance was defined as the vertical displacement of the toe during mid-swing relative to mid-stance, equivalent to the sum of five vertical components: (i) limb shortening resulting from knee flexion, (ii) limb shortening resulting from ankle dorsiflexion, (iii) hip elevation, (iv) contralateral vaulting, and (v) hip abduction (foot circumduction). The latter three components were considered compensatory movements to offset insufficient limb shortening resulting from hemiparesis [23,24,25]. Using this methodology, the total toe clearance and the extent of dependence on compensatory movements were quantitatively evaluated. The contribution of compensatory movements to toe clearance was calculated as a ratio using the following equation:$$Compensatory\;Movements\;Ratio=\frac{Hip\;Elevation+Contralateral\;Vaulting+Hip\;Abduction}{Total\;Toe\;Clearance}$$

### 2.5. Statistical Analyses

Statistical analyses were performed using R software version 4.2.1 (R Foundation for Statistical Computing, Vienna, Austria). Data normality was assessed using the Shapiro–Wilk test. Differences in gait parameters among AFO angle settings were evaluated using repeated-measures ANOVA for normally distributed data and Friedman’s test for non-normally distributed data. Post hoc comparisons were conducted using the paired t-test and the Wilcoxon signed-rank test, with Bonferroni correction being applied for multiple comparisons. Spearman’s rank correlation was used to analyze the relationship between limb shortening and pelvic elevation. Statistical significance was set at *p* < 0.05.

## 3. Results

### 3.1. Participant Characteristics

A total of 26 participants with hemiparetic stroke (mean age: 62.3 ± 14.4 years; 19 male) were included in this study; of these participants, 14 and 12 had right and left hemiparesis, respectively. The median number of days since stroke onset was 55 (interquartile range: 37–621) days. The mean of the Stroke Impairment Assessment Set total lower-limb score [26] (0 = complete motor impairment, 15 = no impairment) was 9.7 ± 3.3, indicating mild-to-moderate impairment in the lower paretic limb. The characteristics of the study participants are summarized in Table 1.

### 3.2. Spatiotemporal Gait Parameters

Spatiotemporal gait parameters are presented according to joint angle settings in Table 2. No significant differences in stride time, paretic swing time, single-stance time, or any stride-length variables were observed among the four AFO joint angles. The only temporal metric that differed under the AFO condition was the double-stance period before paretic toe-off, which shortened significantly with increasing dorsiflexion (*p* < 0.001). Post hoc multiple comparisons indicated that this period was significantly shorter at the 10° (0.37 ± 0.15 s) and 15° (0.37 ± 0.18 s) setting compared to the 0° (0.42 ± 0.20 s; *p* < 0.001 and *p* = 0.002, respectively) and 5° setting (0.40 ± 0.18 s; *p* = 0.011 and *p* = 0.003, respectively).

### 3.3. Joint Angles

The maximal paretic joint angles during each walking phase are presented according to the AFO dorsiflexion angles in Table 3. No significant differences in hip flexion, extension, or knee extension were identified among the four AFO joint angles. Maximal knee flexion significantly decreased as the AFO dorsiflexion angles increased. During the swing phase, it decreased from 33.0 ± 20.0° at 0° to 30.7 ± 19.4° at 10° (*p* = 0.040) and 28.9 ± 19.9° at 15° (*p* = 0.008). Similarly, during the stance phase, maximal knee flexion decreased from 29.4 ± 15.6° at 0° to 27.0 ± 15.1° at 10° (*p* = 0.012) and 24.7 ± 15.5° at 15° (*p* = 0.004). Maximal ankle dorsiflexion during the swing phase significantly increased from −0.8 ± 4.9° at the 0° AFO setting to 1.6 ± 3.6° at 10° (*p* = 0.005) and 1.2 ± 4.5° at 15° (*p* = 0.012); in contrast, no significant change was observed during the stance phase. Ankle plantarflexion during the swing phase decreased from 6.8 ± 5.3° at 0° to 4.0 ± 4.6° at 15° (*p* = 0.014). During the stance phase, an overall effect on ankle plantarflexion was observed (*p* = 0.027), with values decreasing from 6.5 ± 4.4° in the 0° setting to 4.7 ± 4.8° in the 15° setting; however, individual pairwise comparisons were not significant after correction.

### 3.4. Toe Clearance and Its Components

Toe clearance and its components are presented according to the AFO dorsiflexion angles in Table 4. The mean toe clearance during the paretic swing phase was not affected by the AFO dorsiflexion angles (*p* = 0.557); nevertheless, the manner through which clearance was achieved differed. The component of limb shortening resulting from ankle dorsiflexion increased (i.e., became less negative) as AFO dorsiflexion increased (*p* < 0.001), whereas that of compensatory hip elevation decreased (*p* = 0.025). No other compensatory movement components differed between conditions. Limb shortening resulting from knee flexion showed a decreasing trend (*p* = 0.061) but did not reach statistical significance. Pairwise comparisons confirmed a significant increase in the component of limb shortening secondary to ankle dorsiflexion at 5°, 10°, and 15° (−0.8 ± 0.7 cm, −0.7 ± 0.7 cm, and −0.5 ± 0.7 cm), respectively) compared with 0° (−1.1 ± 1.1 cm; *p* < 0.001, *p* = 0.004, and *p* = 0.020, respectively), as well as at 15° compared with 5° (−0.5 ± 0.7 cm vs. −0.8 ± 0.7 cm; *p* = 0.020). Although compensatory hip elevation movement significantly decreased as AFO dorsiflexion increased, post hoc comparisons revealed no significant pairwise differences. The overall compensatory movement ratio declined (*p* = 0.006), driven by reductions at 10° (0.70 ± 0.71 cm) and 15° (0.68 ± 0.76 cm) relative to 5° (0.77 ± 0.75 cm; *p* = 0.005 and *p* = 0.005, respectively). The relative contribution of each component movement to toe clearance is shown in Figure 3.

## 4. Discussion

The present study investigated the influence of varying AFO dorsiflexion angles on gait kinematics in patients with hemiparetic stroke. As the AFO dorsiflexion angle increased, the only spatiotemporal parameter that changed was a decrease in double-stance time before the swing phase. Higher dorsiflexion settings were associated with increased ankle dorsiflexion during the stance and swing phases, as well as reduced knee flexion during the stance phase. Toe clearance analysis revealed that a higher AFO dorsiflexion setting increased ankle-derived limb shortening, which, in turn, reduced compensatory movements such as hip elevation.

Kinematic analyses showed that ankle dorsiflexion during swing increased with higher AFO dorsiflexion angles, whereas plantarflexion decreased during stance and swing. Simultaneously, knee flexion during swing decreased, and no significant changes were observed in the hip motion. These joint-level changes resulted in a shift in the toe clearance strategy: enhanced distal shortening resulting from reduced plantar flexion and a corresponding decline in proximal compensation, especially hip elevation. This may be interpreted as a “trade-off” of energy-intensive proximal compensations for more efficient distal joint mechanics when permitted by the AFO. These results expand on earlier studies demonstrating that AFOs improve toe clearance and reduce hip elevation [11,12,13] by showing the dose-dependent effect of AFO dorsiflexion angles on compensatory movement patterns. The reduction in compensatory movements is particularly relevant because such adaptations, although functionally beneficial, are energetically costly and may contribute to fatigue and long-term musculoskeletal strain [4,10]. The present study demonstrated that even small increases in AFO dorsiflexion (e.g., from 0° to 5°) led to measurable improvements in ankle-limb shortening. This finding highlights the clinical utility of precise AFO tuning for optimizing gait patterns after stroke.

Interestingly, a temporal effect was observed, namely the shortening of the prolonged double-stance phase preceding paretic toe-off with increased dorsiflexion. This temporal improvement aligns with the findings of previous studies on the effects of AFOs [27,28]. Considering the effect of AFOs on limb shortening, a possible mechanism is that increased dorsiflexion reduces peak plantar flexion during the terminal stance, effectively shortening the limb before the swing phase begins. This, in turn, lowers the “clearance demand” placed on the knee and hip, thereby minimizing the need for compensatory movements.

A notable finding was the reduction in knee flexion during the stance phase at higher dorsiflexion settings. This appears counterintuitive, as greater dorsiflexion facilitates tibial progression in mid-stance and can shift the external knee flexion moment [29]. One possible explanation is that an increased AFO dorsiflexion angle promotes distal limb shortening at pre-swing, thereby reducing the toe clearance demand imposed on the knee from pre-swing to early swing, which in turn affects joint angles in the stance and swing phases [30]. The decreased knee flexion during swing may also carry over to stance, reflecting the dynamic interplay between swing and stance phase mechanics [31]. Conversely, considerable inter-individual variability has been reported in the relationship between swing and stance phase knee kinematics, suggesting that additional factors may be involved [32]. Further investigations into stance phase control under varying AFO conditions are warranted.

### 4.1. Clinical Implications

The findings of this study support the individualized adjustment of the AFO dorsiflexion angles as a practical method for reducing inefficient compensatory strategies and enhancing gait smoothness and efficiency. Although increased dorsiflexion did not directly improve toe clearance, it altered the clearance strategy by decreasing reliance on compensatory movements. Clinicians may consider dorsiflexion tuning when addressing toe clearance issues, which often present as delayed swing initiation (reflected in a prolonged pre-swing double-stance time) or excessive compensatory movements during swing—both markers of paretic swing smoothness and efficiency. However, excessive dorsiflexion may limit the need for knee flexion during the swing phase, thereby suppressing dynamic knee flexion and potentially reducing gait adaptability, particularly in uneven and dynamic environments. Based on the graded effects observed, an optimal tuning range of 5–10° is suggested for ambulatory patients with mild-to-moderate impairments. This range offers a favorable trade-off: 5° significantly improved limb shortening, whereas 10° shortened pre-swing double-stance time compared with 0°. Conversely, significant reductions in knee flexion—a potential adverse effect—were observed at 10° and became more pronounced at 15°. Therefore, caution is warranted for patients with stiff-knee gait, as excessive dorsiflexion could exacerbate their gait patterns. AFO tuning should be applied in combination with rehabilitative interventions that promote dynamic knee flexion, using both as complementary strategies to optimize joint coordination and functional mobility.

### 4.2. Limitations

This study had several limitations. First, participants were capable of independent treadmill walking without handrails or orthoses, which limits the generalizability of the findings to patients with more severe impairments. The effects may differ in individuals with greater paresis, given their baseline instability. Second, the mean age of the sample (62.3 years) was relatively young compared with typical stroke demographics, which could have influenced the results. Nevertheless, the observed improvements in toe clearance strategy are mechanistically straightforward and consistent with findings in able-bodied individuals, suggesting this effect is likely robust across impairment levels and age groups. Third, the use of a treadmill may limit direct translation to overground walking. Patients with stroke often show systematic spatiotemporal differences on treadmills—particularly longer double-stance and stance times—but when walking speed is controlled (matched or self-paced), sagittal kinematic waveforms and coordination patterns are largely preserved [33]. Consistent with this, the hip, knee, and ankle angles observed here were within the ranges reported for overground walking with AFOs [34,35]. Thus, although effect sizes may differ overground, the qualitative, angle-dependent shifts in gait strategy are expected to be similar. Fourth, only one type of AFO (the APS-AFO) was tested. Although this design allowed dorsiflexion angle adjustments, other orthotic designs with different mechanical properties may yield different results. Comparative studies using multiple AFO models could broaden clinical applicability. Fifth, this was a single-session study assessing only the immediate kinematic effects of angle adjustments. Long-term adaptation may result in further gait changes as patients habituate to new AFO settings over days or weeks. Finally, electromyographic or detailed kinetic data were not collected, limiting the ability to clarify neuromuscular adaptations and control strategies underlying the observed kinematic changes. Future studies should include such measures to deepen mechanistic understanding.

In summary, this study’s strengths include its within-subject, graded evaluation of AFO dorsiflexion angles and the use of a validated three-dimensional gait analysis system, which together provide concrete clinical suggestions for AFO tuning. The main weaknesses are the treadmill-only design, the absence of kinetic and EMG data, and the relatively young and mildly impaired cohort, which limit the generalizability of the findings. Future research should validate these findings in more diverse settings. Furthermore, the analytical approach could be extended to other daily activities, such as sit-to-stand transitions, toileting, and stair climbing, or even to culturally specific movements now being captured in open-access motion analysis databases [36].

## 5. Conclusions

This study demonstrated that increasing the AFO dorsiflexion angle altered gait kinematics, particularly by improving toe clearance strategies for individuals with hemiparetic stroke without affecting overall gait speed or stride length. These findings suggest that careful adjustment of AFO dorsiflexion angles can enhance gait efficiency and should be considered a part of individualized rehabilitation planning for individuals with hemiparetic stroke.

## Figures and Tables

**Figure 1 bioengineering-12-01091-f001:**
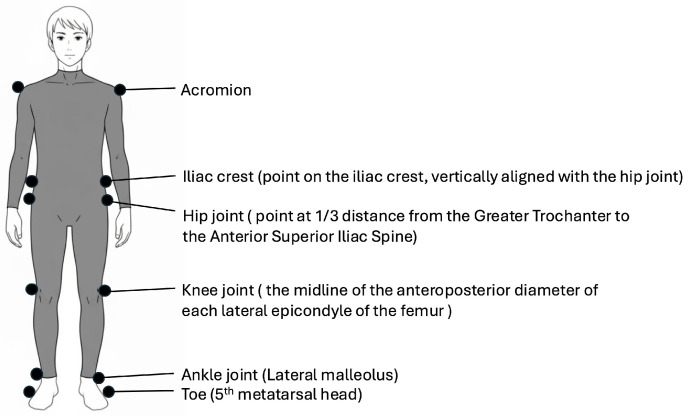
Marker placement. Twelve measurement markers were positioned bilaterally on the acromion, iliac crest, hip, knee, ankle, and toe.

**Figure 2 bioengineering-12-01091-f002:**
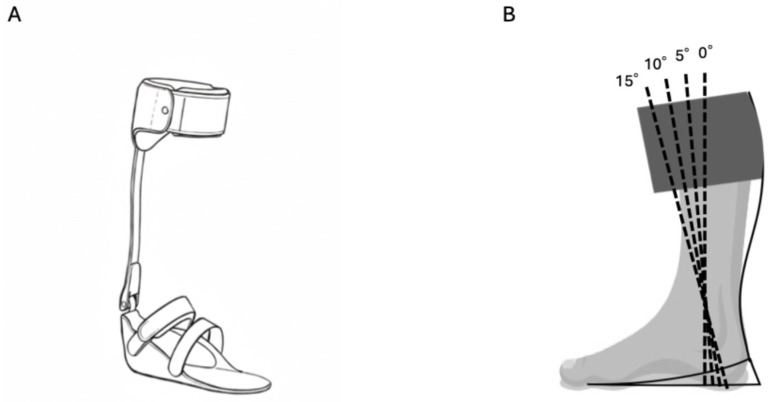
The APS-AFO (adjustable posterior strut ankle-foot orthosis) and angle settings. The APS-AFO consists of a strut (aluminum or carbon) and an ankle hinge joint (**A**). The angle setting is determined by the relationship between the shank and the plantar surface (**B**).

**Figure 3 bioengineering-12-01091-f003:**
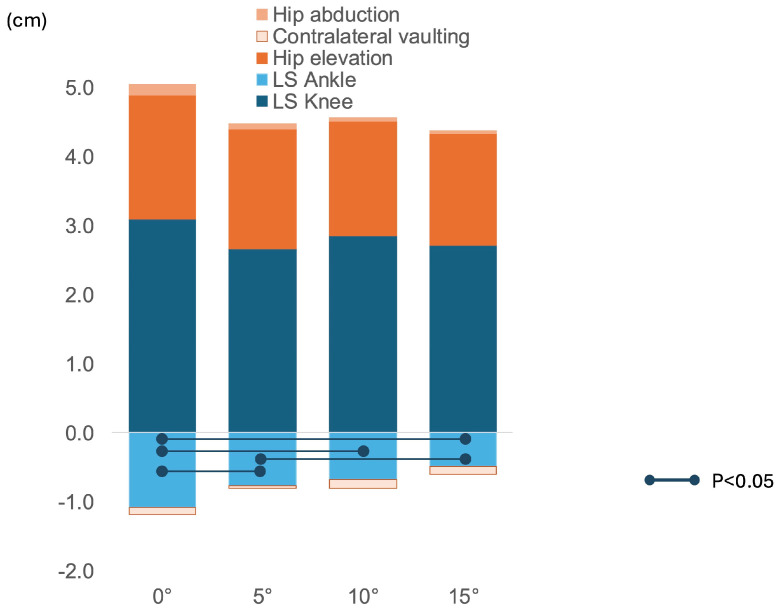
Contribution of individual kinematic components to the toe clearance strategy under different ankle-foot orthosis dorsiflexion angles. The stacked bars illustrate the vertical components (in centimeters) of limb shortening (LS Knee, LS Ankle) and compensatory movements (hip elevation, contralateral vaulting, and hip abduction), which collectively contribute to toe clearance. Positive values on the *y*-axis represent the contribution to increased toe clearance, whereas negative values represent the contribution to decreased toe clearance. The sum of these components equals the net toe clearance. The symbol “●” indicates a statistically significant difference (*p* < 0.05) for that component across the angular conditions, as determined by pairwise comparisons. LS Knee: limb shortening resulting from knee flexion; LS Ankle: limb shortening resulting from ankle dorsiflexion.

**Table 1 bioengineering-12-01091-t001:** Participants’ characteristics (*N* = 26).

Characteristics	
Age, years	62.3 ± 14.4
Sex, male/female	19/7
Type of stroke, hemorrhagic/ischemic	14/12
Affected side, right/left	14/12
Stroke Impairment Assessment Set total lower-limb score	9.7 ± 3.3
Days after onset	55 (37–621)
Functional Independence Measure-Walking score, 4/5/6	6/9/11

Values are presented as numbers, means ± standard deviations, or medians (interquartile ranges).

**Table 2 bioengineering-12-01091-t002:** Spatiotemporal gait parameters according to each AFO dorsiflexion angle.

Variable	AFO Dorsiflexion Angles				*p* Value ^†^
	0°	5°	10°	15°	
Stride length, cm	59.3 ± 30.5	57.8 ± 28.5	57.8 ± 28.4	58.3 ± 28.2	0.956
Paretic step length, cm	30.1 ± 15.5	29.3 ± 14.7	29.0 ± 15.0	28.6 ± 15.0	0.485
Non-paretic step length, cm	29.2 ± 16.5	28.5 ± 15.3	28.8 ± 14.8	29.7 ± 15.0	0.557
Stride time, s	1.70 ± 0.47	1.68 ± 0.44	1.67 ± 0.45	1.66 ± 0.51	0.151
Paretic swing time, s	0.49 ± 0.14	0.49 ± 0.15	0.51 ± 0.19	0.49 ± 0.13	0.406
Paretic single-stance time, s	1.20 ± 0.41	1.19 ± 0.34	1.16 ± 0.34	1.18 ± 0.43	0.105
Double-stance time after paretic swing, s	0.38 ± 0.21	0.38 ± 0.15	0.38 ± 0.17	0.40 ± 0.24	0.172
Double-stance time before paretic swing, s	0.42 ± 0.20	0.40 ± 0.18	0.37 ± 0.15	0.37 ± 0.18	<0.001

Values are presented as means ± standard deviations. ^†^ Friedman test across four AFO dorsiflexion angles. Significant pairs with post hoc pairwise comparisons: double-stance time before the paretic swing, 0° vs. 10° (*p* < 0.001), 0° vs. 15° (*p* = 0.002), 5° vs. 10° (*p* = 0.011), and 5° vs. 15° (*p* = 0.003). AFO, ankle-foot orthosis.

**Table 3 bioengineering-12-01091-t003:** Maximal paretic joint angles during each walking phase according to each AFO dorsiflexion angle.

Angles, Degree		AFO Dorsiflexion Angles				*p* Value ^†^
		0°	5°	10°	15°	
Hip flexion	Swing	25.4 ± 9.0	23.8 ± 8.5	25.4 ± 8.2	25.4 ± 9.2	0.227
	Stance	20.4 ± 8.3	21.8 ± 8.0	22.3 ± 7.9	21.8 ± 8.8	0.485
Hip extension	Swing	−9.1 ± 9.5	−9.2 ± 9.8	−8.5 ± 9.5	−8.1 ± 9.3	0.934
	Stance	1.8 ± 8.1	1.2 ± 9.4	1.5 ± 8.3	2.1 ± 8.4	0.875
Knee flexion	Swing	33.0 ± 20.0	31.4 ± 19.7	30.7 ± 19.4	28.9 ± 19.9	0.023
	Stance	29.4 ± 15.6	27.3 ± 15.2	27.0 ± 15.1	24.7 ± 15.5	0.022
Knee extension	Swing	3.6 ± 9.1	4.1 ± 9.0	3.9 ± 8.3	3.4 ± 9.1	0.853
	Stance	−3.5 ± 6.6	−2.8 ± 7.0	−2.8 ± 5.9	−3.2 ± 6.8	0.731
Ankle dorsiflexion	Swing	−0.8 ± 4.9	−0.5 ± 3.8	1.6 ± 3.6	1.2 ± 4.5	<0.001
	Stance	4.9 ± 5.3	4.3 ± 3.7	5.3 ± 3.3	5.0 ± 4.6	0.145
Ankle plantarflexion	Swing	6.8 ± 5.3	5.7 ± 3.8	4.3 ± 5.0	4.0 ± 4.6	0.001
	Stance	6.5 ± 4.4	5.5 ± 4.0	5.0 ± 4.6	4.7 ± 4.8	0.027

Values are presented as means ± standard deviations. ^†^ Friedman test across four AFO dorsiflexion angles. Significant pairs with post hoc pairwise comparisons: swing knee flexion, 0° vs. 15° (*p* = 0.008) and 0° vs. 10° (*p* = 0.040); swing ankle dorsiflexion, 0° vs. 15° (*p* = 0.012) and 0° vs. 10° (*p* = 0.005); swing ankle plantarflexion, 0° vs. 15° (*p* = 0.014); stance knee flexion, 0° vs. 15° (*p* = 0.004) and 0° vs. 10° (*p* = 0.012). AFO, ankle-foot orthosis.

**Table 4 bioengineering-12-01091-t004:** Toe clearance and its components according to AFO dorsiflexion angles.

Variable	AFO Dorsiflexion Angles				*p* Value ^†^
	0°	5°	10°	15°	
Toe clearance, cm	3.9 ± 2.0	3.7 ± 1.9	3.8 ± 2.1	3.8 ± 2.1	0.557
*Toe clearance components*					
Limb shortening secondary to knee flexion, cm	3.1 ± 3.2	2.7 ± 3.2	2.9 ± 3.3	2.7 ± 3.3	0.061
Limb shortening secondary to ankle dorsiflexion, cm	−1.1 ± 1.1	−0.8 ± 0.7	−0.7 ± 0.7	−0.5 ± 0.7	<0.001
Compensatory hip elevation, cm	1.8 ± 1.7	1.7 ± 1.8	1.7 ± 1.8	1.6 ± 1.8	0.025
Compensatory contralateral vaulting, cm	−0.1 ± 0.9	−0.0 ± 1.0	−0.1 ± 1.1	−0.1 ± 1.0	0.831
Compensatory hip abduction, cm	0.2 ± 0.4	0.1 ± 0.3	0.1 ± 0.3	0.1 ± 0.3	0.163
Compensatory movement ratio	0.74 ± 0.73	0.77 ± 0.75	0.70 ± 0.71	0.68 ± 0.76	0.006

Values are presented as means ± standard deviations. ^†^ Friedman test across four AFO dorsiflexion angles. Significant pairs with post hoc pairwise comparisons: ankle component of limb shortening, 0° vs. 5° (*p* = 0.020), 0° vs. 10° (*p* = 0.004), 0° vs. 15° (*p <* 0.001), and 5° vs. 15° (*p* = 0.020); and compensatory movement ratio, 0° vs. 15° (*p* = 0.005) and 5° vs. 15° (*p* = 0.005). AFO, ankle-foot orthosis.

## Data Availability

Datasets used and/or analyzed during the current study are available from the corresponding author upon reasonable request. The data are not publicly available due to ethical and privacy concerns.

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
