# Peer review of "Effect of the Ankle–Foot Orthosis Dorsiflexion Angle on Gait Kinematics in Individuals with Hemiparetic Stroke"

_bioengineering, 2025, doi:10.3390/bioengineering12101091_

Round 1
Reviewer 1 Report
Comments and Suggestions for Authors
The manuscript entitled “Effect of the Ankle-Foot Orthosis Dorsiflexion Angle on Gait Kinematics in Individuals with Hemiparetic Stroke” addresses an important and clinically relevant topic in post-stroke gait rehabilitation. The study is sound, but there are some issues that might improve.
The results show a significant decrease in knee flexion with increasing AFO dorsiflexion angles. The discussion interprets this finding as possibly due to a dynamic interplay between swing and stance phases. However, this explanation remains somewhat speculative. I encourage the authors to expand this section with additional biomechanical reasoning or reference to prior literature, as this point is central to understanding the trade-offs of AFO adjustment.
The participants were able to walk independently on a treadmill, which may exclude a large proportion of stroke survivors with more severe gait impairments. It would strengthen the paper to explicitly state the degree to which these findings can—or cannot—be extrapolated to less functional populations, and to discuss whether similar effects could be expected in overground walking.
While the manuscript concludes that “careful adjustment” of dorsiflexion angles is beneficial, clinicians may benefit from more concrete guidance. For instance, is there an optimal range (e.g., 5–10°) that appears most advantageous in balancing compensatory movements and knee flexion requirements? More specific recommendations, even if preliminary, would enhance the clinical impact.
There are also some minor issues that can be corrected
-
Original: “This study was conducted in accordance with the ethical principles embodied in the 1964 Declaration of Helsinki, as revised in 2013.”
Correction: “This study was conducted in accordance with the principles of the Declaration of Helsinki (1964), as revised in 2013.” -
Original: “...the order of measurements was randomized, with a rest period of 5–10 min between sessions.”
Correction: “...the order of measurements was randomized, and participants rested for 5–10 minutes between sessions.” -
Original: “Values are presented as numbers, means (standard deviations), or medians (interquartile range).”
Correction: “Values are presented as numbers, means ± standard deviations, or medians (interquartile ranges).” -
Original: “This study demonstrated that increasing the AFO dorsiflexion angles altered the gait kinematics, particularly improving toe clearance strategies...”
Correction: “This study demonstrated that increasing the AFO dorsiflexion angle altered gait kinematics, particularly by improving toe clearance strategies...”
Reviewer 2 Report
Comments and Suggestions for Authors
The study addresses an important clinical question: how systematic adjustment of AFO dorsiflexion angles influences gait kinematics in post-stroke patients. The work is methodologically rigorous, clinically relevant, and adds value to existing literature. However, the paper would benefit from stronger contextual framing, a deeper biomechanical interpretation of key findings (especially reduced knee flexion), and more explicit discussion of clinical applicability.
- Introduction and Literature Gap
- The introduction acknowledges prior work but does not clearly differentiate from studies in healthy adults (Hosokawa et al. 2024) or single-setting stroke studies (Pongpipatpaiboon et al. 2018).
🔹 Suggestion: Explicitly highlight novelty—this is the first stroke study with graded dorsiflexion AFO adjustments.
- Methods
- Treadmill-based gait may not generalize to overground walking. Justification is needed. Provide evidence to norm data, for example as in Iso standards of the hip , knee , ankle angles of implant tests etc or some normative database data versus treadmill data .
Simultaneous validation of wearable motion capture system for lower body applications: over single plane range of motion (ROM) and gait activities, BIOMEDICAL ENGINEERING-BIOMEDIZINISCHE TECHNIK, 2022, 0013-5585, 67, 3, 185-199
- Participant selection: only independently ambulatory patients were included; findings may not apply to severely impaired patients.
- Biomechanical mechanisms: No EMG or kinetic data to explain reduced knee flexion.
🔹 Suggestion: State this clearly and recommend future multimodal assessments.
- Results
- Clinical significance is under-discussed. For instance, shortening double stance by ~0.05 s — is this meaningful for fall risk or fatigue reduction?
- Toe clearance height unchanged, but strategies altered. This distinction should be emphasized for clinicians.
- Discussion
- The explanation of reduced knee flexion is speculative. Other possible mechanisms (altered tibial progression, quadriceps activity, stance-swing coupling) should be discussed.
- Statements about energy efficiency are inferred, not measured. Authors should clarify this is a hypothesis.
- Clinical recommendations should be more concrete, e.g., “settings of 10–15° may be optimal for patients with mild hemiparesis, but caution is needed for those with stiff-knee gait.”
- You could provide a SWOT analysis of your study and for future perspectives, if your methodology could be applied to activities of daily life such as in literature below Database covering the prayer movements which were not available previously, NATURE SCIENTIFIC DATA, 2023, 2052-4463, 10, 1. v
- Limitations
- Well-written, but should stress:
- Single-session study, no adaptation effects.
- Relatively young sample compared with typical stroke demographics.
- One AFO type tested (APS-AFO); results may differ with solid or hinged designs.
- Presentation
- Figures are clear, but a schematic of the AFO setup and marker placement would aid readers.
- Ensure consistent terminology (“double-stance” vs. “double-support”).
- Minor English style refinements would improve readability.
Specific Recommendations
- Strengthen Introduction with explicit contrast to previous studies.
- Expand on clinical significance of findings (effect sizes, patient implications).
- Provide deeper biomechanical reasoning for reduced knee flexion.
- Clarify speculative vs. proven outcomes (e.g., energy cost).
- Recommend multimodal future studies (kinetics, EMG).
- Add schematic figures and improve consistency of terminology.
could be revised
Round 2
Reviewer 1 Report
Comments and Suggestions for Authors
Thank you to improve the paper following my suggestions